# The Role of GPR109a Signaling in Niacin Induced Effects on Fed and Fasted Hepatic Metabolism

**DOI:** 10.3390/ijms22084001

**Published:** 2021-04-13

**Authors:** Caroline E. Geisler, Kendra E. Miller, Susma Ghimire, Benjamin J. Renquist

**Affiliations:** 1School of Animal and Comparative Biomedical Sciences, University of Arizona, Tucson, AZ 85721, USA; caroline.geisler@pennmedicine.upenn.edu (C.E.G.); kendramiller@arizona.edu (K.E.M.); ghimires@email.arizona.edu (S.G.); 2Perelman School of Medicine, University of Pennsylvania, Philadelphia, PA 19104, USA

**Keywords:** GPR109a, β-OH butyrate, niacin, metabolic homeostasis, liver

## Abstract

Signaling through GPR109a, the putative receptor for the endogenous ligand β-OH butyrate, inhibits adipose tissue lipolysis. Niacin, an anti-atherosclerotic drug that can induce insulin resistance, activates GPR109a at nM concentrations. GPR109a is not essential for niacin to improve serum lipid profiles. To better understand the involvement of GPR109a signaling in regulating glucose and lipid metabolism, we treated GPR109a wild-type (+/+) and knockout (−/−) mice with repeated overnight injections of saline or niacin in physiological states characterized by low (ad libitum fed) or high (16 h fasted) concentrations of the endogenous ligand, β-OH butyrate. In the fed state, niacin increased expression of apolipoprotein-A1 mRNA and decreased sterol regulatory element-binding protein 1 mRNA independent of genotype, suggesting a possible GPR109a independent mechanism by which niacin increases high-density lipoprotein (HDL) production and limits transcriptional upregulation of lipogenic genes. Niacin decreased fasting serum non-esterified fatty acid concentrations in both GPR109a +/+ and −/− mice. Independent of GPR109a expression, niacin blunted fast-induced hepatic triglyceride accumulation and peroxisome proliferator-activated receptor α mRNA expression. Although unaffected by niacin treatment, fasting serum HDL concentrations were lower in GPR109a knockout mice. Surprisingly, GPR109a knockout did not affect glucose or lipid homeostasis or hepatic gene expression in either fed or fasted mice. In turn, GPR109a does not appear to be essential for the metabolic response to the fasting ketogenic state or the acute effects of niacin.

## 1. Introduction

GPR109a was identified as a niacin receptor in 2003 [1]. Although niacin binds to GPR109a with high affinity (100 nM EC_50_), this concentration is only reached in response to the administration of pharmacological doses. In 2005, it was established that physiologically relevant concentrations of β-OH butyrate activated GPR109a [2]. With an EC_50_ of 700–800 µM, physiologically relevant changes in β-OH butyrate concentrations that accompany a fast can activate GPR109a signaling [3]. While GPR109a was first shown to inhibit adipose tissue lipolysis, it has since been identified in various other tissues with a broad range of physiological actions [2]. The GPR109a agonist, niacin, regulates gene expression in the liver, skeletal muscle, adipose tissue, and macrophages, although a direct role of GPR109a signaling has not been explored [4,5,6,7].

Niacin is a powerful anti-atherosclerotic lipid-lowering drug whose clinical potential was first recognized over half a century ago [8]. Niacin effectively decreases circulating triglyceride (TAG) and low-density lipoprotein (VLDL) concentrations while raising high-density lipoprotein (HDL) levels in patients with dyslipidemia [9,10]. While statins became the dominant therapy for hypercholesterolemia after their introduction in 1987, niacin is prescribed in statin-resistant individuals, and the benefits of statin/niacin combination treatment are under debate [11,12,13]. Interestingly, niacin was found to improve plasma cholesterol levels in GPR109a −/− mice, questioning the underlying role of GPR109a in niacin’s lipid efficacy [14]. Studies investigating GPR109a dependent and independent components of niacin signaling are necessary to maximize the clinical applications of niacin therapy. The flushing response to niacin is GPR109a mediated and is a primary reason that patients discontinue niacin therapy [15,16]. Better understanding the GPR109a dependent and independent effects of niacin will inform the development of niacin mimetics that confer the beneficial metabolic effects but lack the flushing response.

Both obesity and fasting are ketotic states, and given the rising interest in intermittent fasting and the rising incidence of obesity, it is essential that we gain a better understanding of the metabolic consequences of GPR109a signaling. Using HMGCS2 knockdown, we previously established that ketones were important regulators of the metabolic response to a fast [3]. In the studies presented here, we expand upon those findings to focus on the role of GPR109a in this metabolic feedback. These studies focused on glucose, lipid, and cholesterol homeostasis, hepatic metabolic enzyme mRNA expression, and serum lipid, cholesterol, and ketogenic profiles, allowing us to assess the role of whole-body GPR109a in the normal fasting response and pharmacological effects of niacin.

## 2. Results

Global GPR109a knockout was confirmed by quantifying GPR109a mRNA expression in liver and white adipose tissue (Appendix A). We first investigated the metabolic response to niacin treatment in fed state wild-type (WT) and GPR109a null mice. Niacin decreased hepatic glycogen content in both genotypes but did not alter serum glucose concentrations (Figure 1A,B). Niacin did not affect serum insulin or the glucose:insulin ratio (Figure 1C,D). Additionally, glucose clearance during an IP glucose tolerance test and glucose-stimulated serum insulin concentrations were not different between GPR109a +/+ and −/− mice (Figure 1E–G). As niacin modulates cholesterol and triglyceride metabolism [8,9], we assessed the lipid and cholesterol profiles in niacin-treated GPR109a +/+ and −/− mice. Acute niacin treatment had no effect on serum or hepatic non-esterified fatty acid (NEFA) and triglyceride (TAG) concentrations in the fed state (Figure 2A–D). Serum β-OH butyrate concentrations were not affected by niacin in either genotype (Figure 2E). Additionally, niacin treatment did not alter serum low-density lipoprotein (LDL) or high-density lipoprotein (HDL) concentrations (Figure 2F,G).

We expected that GPR109a signaling might exert physiologically relevant regulation of mRNA expression of enzymes in metabolic pathways that are active when production of the endogenous GPR109a ligand, β-OH butyrate, is upregulated [2]. Accordingly, we examined hepatic mRNA expression of key genes in gluconeogenesis, β-oxidation, and ketogenesis. Fed state hepatic mRNA expression of the lipid activated transcription factor PPARα [17] was not affected by niacin treatment in either genotype (Figure 3A). Niacin doubled fed hepatic mRNA expression of the early gluconeogenic gene, phosphoenolpyruvate carboxykinase (PEPCK), in GPR109a null mice (Figure 3B). The mitochondrial uncoupling protein 2 (UCP2) is essential for NAD^+^ regeneration during fasting to support high rates of lipid oxidation and ketone production [18,19]. Niacin did not alter fed hepatic UCP2 expression independent of GPR109a expression (Figure 3C). Niacin also did not alter fed hepatic mRNA expression of the rate-limiting enzyme in ketogenesis, hydroxy-methylglutaryl-CoA synthase 2 (HMGCS2), in either genotype (Figure 3D). Hepatic mRNA expression of the mitochondrial long-chain fatty acid transporter that regulates lipid entry to β-oxidation, carnitine palmitoyltransferase 1 (CPT1), was increased by niacin in GPR109a −/− but not +/+ mice (Figure 3E). However, niacin treatment had no impact on protein expression of hepatic CPT1 in either genotype (Figure 3F).

As niacin therapy elevates HDL and lowers VLDL and TAG concentrations in humans [9,10], we assessed the impact of niacin treatment on the mRNA expression of key hepatic genes regulating lipogenesis and cholesterol synthesis. Niacin did not alter fed state hepatic mRNA expression of the cytosolic enzyme hydroxy-methylglutaryl-CoA synthase 1 (HMGCS1), which generates the precursor for cholesterol synthesis, or hydroxy-methylglutaryl-CoA reductase (HMGCR), the rate-controlling enzyme in the mevalonate pathway (Figure 3G,H). Hepatic mRNA expression of apolipoprotein A1 (Apo-A1), the dominant protein component of HDL particles [20], was elevated by niacin treatment independent of genotype (Figure 3I; *p* = 0.009). However, this was not significant within GPR109a +/+ or GPR109a −/− individually. Sterol regulatory element-binding protein 1 (SREBP1) is an important transcriptional promoter of lipogenesis that is activated by insulin signaling in the fed state [21]. Niacin decreased fed state hepatic SREBP1 mRNA expression independent of GPR109a expression (Figure 3J; *p* = 0.029), although this was not significant within either genotype. There was no effect of niacin on fed state hepatic mRNA expression of SREBP2, a transcription factor that promotes the expression of cholesterogenic genes, including HMGCR [22].

The lack of a robust metabolic phenotype in response to niacin treatment in the fed state, independent of GPR109a expression, prompted us to next examine the effect of repeated niacin injections on hepatic metabolic homeostasis during a 16 h fast. Niacin decreased fasted hepatic glycogen concentrations only in GPR109a −/− mice and did not affect serum glucose, insulin, or the glucose:insulin ratio in either genotype (Figure 4A–D). We report that repeated niacin injections during a 16 h fast decreased serum NEFA and TAG concentrations in both GPR109a +/+ and −/− mice (Figure 5A,B). Niacin tended to decrease fasted hepatic NEFA concentrations in GPR109a null mice but had no effect in wild-type mice (Figure 5C; *p* < 0.1). Niacin decreased liver TAG concentrations by ~25% in both genotypes (Figure 5D). Additionally, niacin treatment diminished serum β-OH butyrate concentrations independent of genotype (Figure 5E; *p* = 0.0046). Although this only reached significance in wild-type mice (22% decrease), niacin also decreased serum β-OH butyrate by 16% in GPR109a −/− mice (Figure 5E). Interestingly, niacin decreased fasted serum LDL concentrations only in GPR109a null mice (Figure 5F). While there was no effect of niacin on fasted serum HDL concentrations, GPR109a null mice had significantly lower HDL concentrations independent of injection treatment (Figure 5G; *p* = 0.0025).

Niacin decreased hepatic PPARα mRNA expression in fasted mice independent of GPR109a expression (Figure 6A; *p* = 0.017). However, this decrease was only significant in wild-type mice (Figure 6A). Despite the muted expression of PPARα mRNA with niacin treatment, expression of mRNA for the PPARα target genes UCP2, CPT1, HMGCS2, and PEPCK [23,24,25,26,27] was not altered by niacin treatment in either genotype in fasting mice (Figure 6B–E). Fasted hepatic CPT1 protein expression was also not altered by niacin treatment (Figure 6F). Interestingly, hepatic HMGCS1 mRNA expression was elevated in GPR109a null mice independent of injection treatment (*p* = 0.0031), while niacin had no effect in either genotype (Figure 6G). Niacin treatment did not affect fasted hepatic mRNA expression of HMGCR, Apo-A1, SREBP1, or SREBP2 (Figure 6H–K). Still, SREBP1 mRNA tended to be elevated in WT saline-treated mice relative to GPR109a null mice (*p* = 0.07).

## 3. Discussion

Niacin has been used as a broad-spectrum lipid-lowering drug for over 60 years [8]. Although renowned clinically for its anti-atherosclerotic properties, niacin affects whole-body glucose and lipid homeostasis. Niacin’s mechanism of action has been under investigation since it was first used in the clinic, and recent research continues to reveal new complexities. We investigated the role of GPR109a expression in regulating serum and hepatic metabolites and hepatic gene expression in the fed and fasted state in response to repeated overnight niacin or saline injections.

Niacin induces insulin resistance and fasting hyperglycemia [28,29]. In fact, niacin treatment for as little as one week decreases insulin-stimulated glucose clearance [30]. Niacin could cause insulin resistance by altering skeletal muscle glucose utilization. Niacin increases the number of oxidative type 1 skeletal muscle fibers, a phenotype that favors lipid over glucose oxidation [6,31]. This decrease in glycolytic fibers may diminish muscle glucose utilization and impair insulin sensitivity. In support, niacin-mediated increases in muscle lipid oxidation were correlated with niacin-induced decreases in insulin sensitivity [32]. Although acute niacin signaling at adipocyte GPR109a inhibits lipolysis and decreases circulating NEFA concentrations, sustained niacin treatment causes NEFA levels to rebound to or above basal concentrations [1,33]. This NEFA rebound has been implicated in niacin-induced insulin resistance and may provide the substrate necessary to limit skeletal muscle glucose demand [34,35,36]. We report that GPR109a knockout did not alter glucose tolerance (Figure 1E). Additionally, while niacin has been shown to decrease glucose-stimulated insulin secretion through a GPR109a dependent mechanism [37,38], we found no effect of GPR109a knockout on glucose-stimulated serum insulin concentrations (Figure 1G). While these results do not negate a role for GPR109a signaling in niacin-induced insulin resistance, they suggest that endogenous GPR109a signaling does not affect glucose tolerance or glucose-stimulated serum insulin. As the endogenous GPR109a ligand, β-OH butyrate is at subthreshold concentrations to activate GPR109a during the fed state, GPR109a signaling is not physiologically engaged post-prandially, and thus, the absence of GPR109a expression would not be expected to alter glucose-stimulated insulin release or glucose clearance. In support of this, basal insulin and glucose-stimulated insulin secretion were not different between vehicle-treated GPR109a knockdown and control INS-1 cells [37].

Niacin treatment robustly decreased fed state hepatic glycogen concentrations independent of GPR109a expression (Figure 1A). This is consistent with evidence that niacin decreases hepatic glycogen in mammalian and avian species [39,40]. Three weeks of dietary niacin supplementation increased hepatic glycogen phosphorylase activity in basal fed, 48 h fasted, and 24 h refed turkey poults with no change in glycogen synthase activity [40]. This suggests niacin decreases glycogen concentrations by increasing glycogenolysis. Glycogen phosphorylase activity is negatively regulated by acetylation, and SIRT1 increases glycogen phosphorylase activity [41]. Niacin is a substrate for NAD^+^ synthesis, and NAD^+^-dependent activation of the deacetylase SIRT1 has been proposed to mediate some of niacin’s effects [42,43,44]. However, repeated overnight injections of niacin but not nicotinamide, another NAD^+^ precursor, decreased liver glycogen concentrations in rats [39,43]. Thus, it appears that niacin decreases hepatic glycogen stores independent of NAD^+^ concentration or GPR109a signaling. Interestingly, in the fasted state, the niacin-induced decrease in glycogen was only evident in GPR109a knockout mice (Figure 5A). Although significant, the physiological impact of this finding is likely minimal, as hepatic glycogen stores are almost entirely depleted following a 16 h fast [3].

Niacin potently improves lipid metabolism by decreasing triglyceride, VLDL, and LDL concentrations and increasing HDL concentrations [9,14,45]. Central to the long-standing free fatty acid hypothesis explanation for niacin’s favorable lipoprotein effects is the notion that adipose-derived NEFAs taken up by the liver can be re-esterified into TAGs which can then be incorporated into VLDL particles [46]. Accordingly, it was believed that niacin’s action to inhibit adipose lipolysis resulted in decreased TAG and VLDL production by limiting substrate availability [47]. However, more recent findings have established that niacin acts through several mechanisms directly at the liver, which decrease VLDL and increase HDL concentrations, questioning the validity of the free fatty acid hypothesis [48,49,50,51,52]. Moreover, it has been shown that while niacin’s anti-lipolytic effects are GPR109a dependent, niacin still decreases plasma TAG and VLDL and increases HDL concentrations in GPR109a −/− mice [14]. Interestingly, we show that fasted GPR109a null mice have decreased serum HDL concentrations independent of injection treatment (Figure 5G), suggesting that although GPR109a has been shown not to be required for niacin to increase HDL [14], GPR109a signaling may still regulate circulating HDL concentrations during a fast.

We report that niacin decreased fasting serum NEFA and TAG concentrations in both GPR109a +/+ and −/− mice (Figure 5A,B). This is in direct contrast to previous findings that niacin does not decrease plasma-free fatty acids in GPR109a null mice [1,14]. One possible explanation for this discrepancy is the timing between niacin exposure and NEFA quantification. In studies that showed GPR109a dependent niacin responses, plasma-free fatty acids were measured within 60 min or less of niacin administration [1,14]. Supporting this acute vs. chronic response, Lauring et al. (2012) presented an anti-lipolytic effect of niacin within 15 min that was GPR109a dependent and a decrease in pro-atherosclerotic plasma TAG and VLDL after four days of niacin treatment that was GPR109a independent [14]. We assessed serum NEFAs 1–2 h after the last niacin injection and following 9 h of repeated niacin injections. To the best of our knowledge, we are the first to report serum-free fatty acid concentrations in GPR109a null mice treated with niacin for more than one hour. It is also possible that differences in niacin dose, route of administration, and frequency of treatment could contribute to the reported differing effect of niacin on serum NEFA concentrations. We failed to observe the expected changes in circulating cholesterol particles following niacin treatment apart from lower fasting serum LDL in niacin-treated GPR109a −/− mice (Figure 5F). It is possible that more chronic niacin exposure is required to reflect niacin-induced changes in serum cholesterol. Indeed, the shortest duration of niacin treatment in the mouse after which cholesterol concentrations have been reported is four days [14]. Future studies assessing the time course of changing lipid and cholesterol profiles following niacin treatment may be pertinent.

Supporting a physiological relevance of our findings, GPR109a expression did not affect the elevation in serum NEFA in response to a fast. If GPR109a was a significant regulator of serum NEFA, one would expect that fasting would result in a greater rise in serum NEFA, liver NEFA, liver TAG, or serum TAG in GPR109a null mice. It remains possible that niacin lowers serum NEFA concentrations by increasing non-hepatic NEFA clearance. However, an increase in NEFA clearance would be observable in the fed and fasted state, while the niacin-induced decrease in serum NEFAs was specific to the fasted state. The mechanism by which niacin inhibits adipose tissue lipolysis and NEFA release through a GPR109a independent mechanism warrants further investigation and may hold the key to developing lipid-lowering niacin mimetic that does not result in flushing. Diminished circulating NEFAs, which provide substrate for hepatic TAG and ketone synthesis, likely explain the niacin-induced decrease in liver TAG and serum β-OH butyrate concentrations (Figure 5D,E). However, niacin can decrease TAG production by inhibiting diacylglycerol acyltransferase 2 (DGAT2) activity [49].

Niacin regulates gene expression and alters the expression of lipoprotein transporters and receptors, accounting for some of niacin’s anti-atherosclerotic effects [48,51,52,53]. Niacin has been reported to increase apo-A1 gene expression in HepG2 cells [54], and we extend these findings to show an effect of niacin to increase fed state hepatic apo-A1 mRNA expression in the mouse (Figure 3I). This may underlie the increase in apo-A1 production and HDL concentrations observed with niacin therapy [55]. Niacin treatment also decreases both VLDL number and particle size, and the decrease in VLDL size is due to decreased VLDL TAG content [9]. We observed a treatment effect of niacin to decrease fed state hepatic mRNA expression of the lipogenic transcription factor SREBP1 (Figure 3J). Inhibiting hepatic lipogenic gene expression, downstream TAG synthesis, and VLDL-TAG packaging is likely one mechanism underlying niacin’s cholesterol-lowering effects. Indeed, others report that niacin diminishes SREBP1 mRNA and protein expression; however, this has been proposed to both be dependent on and independent of GPR109a expression [56,57].

The mechanism for how niacin regulates gene expression is not well understood. One hypothesis is that niacin exerts cAMP and liver x receptor α (LXRα) dependent transcriptional regulation, which is proposed to be indirectly downstream of GPR109a activation [5,48,51,53,58,59]. We observed that niacin upregulated fed state hepatic CPT1 and PEPCK mRNA expression in GPR109a −/− mice (Figure 3B,E). One possible GPR109a independent mechanism of niacin-regulated gene expression is through NAD^+^ mediated activation of SIRT1 [44]. SIRT1 activates the transcriptional coactivator peroxisome proliferator-activated receptor γ-coactivator 1α (PGC-1α), and PGC-1α upregulates hepatic expression of PEPCK and CPT1 [60,61,62]. This does not explain why niacin-mediated upregulation of CPT1 was only evident in the absence of GPR109a signaling. One might hypothesize that GPR109a signaling decreases the expression of CPT1, while niacin’s GPR109a independent signaling increases the expression of CPT1. Accordingly, these opposing effects of niacin on CPT1 mRNA expression offset in wild-type mice. CPT1 mRNA expression was 42% greater in fasted GPR109a null mice than in WT mice, supporting a negative feedback role for CPT1 (Figure 6E; genotype effect *p* = 0.01). However, this increase in CPT1 mRNA expression with niacin did not translate into elevated CPT1 protein expression (Figure 6F).

PPARα is a master regulator of hepatic fasting metabolism, which coordinates upregulation of gluconeogenesis, β-oxidation, and ketogenesis [24]. Hepatic PPARα is activated by unsaturated fatty acids and upregulates expression of itself through PPARα response elements in its promoter [17,63]. A decreased supply of NEFAs to the liver could explain the blunted fasting PPARα expression with niacin treatment (Figure 6A). Surprisingly, despite the niacin induced 47% reduction in fasted PPARα expression in wild-type mice and 23% reduction in GPR109a null mice, expression of the PPARα target genes UCP2, CPT1, HMGCS2, and PEPCK [23,24,25,26,27] were unaffected by niacin treatment (Figure 6).

## 4. Materials and Methods

### 4.1. Animals

All studies were conducted using 12–14 week old male whole-body GPR109a +/+ or −/− littermates derived from in-house crosses of GPR109a +/− mice. The founding GPR109a −/− mice were kindly provided by Klaus Pfeffer at the Institute of Medical Microbiology, Immunology, and Hygiene at Heinrich Heine University [1]. Mice were kept on a 14 h light/10 h dark schedule and housed 3–4 mice per cage until 1 week prior to study initiation, at which point animals were individually housed. Ad libitum access to NIH-31 chow (Harlan Laboratories, Indianapolis, IN, USA) and water was available. All studies were approved by the University of Arizona Institutional Animal Care and Use Committee Protocol 13-456 (Approved 21 July 2016).

### 4.2. Injection Studies

Mice were singly housed one week prior to experimentation. 16 h before sacrifice, all mice were switched to Sani-Chip bedding (Harlan Laboratories, Indianapolis, IN, USA; Cat. # 7090 Sani-Chips), and food was removed from mice in the fasted group. All mice had ad libitum access to water. Intraperitoneal injections of 0.8 mmol/kg (approximately 100 mg/kg) GPR109a agonist nicotinic acid (niacin) or phosphate-buffered saline (PBS) were given at 0.1 mL/10 g body weight 9, 7, 5, 3, and 1 h before sacrifice. This dose has been previously shown to limit plasma non-esterified fatty acids for at least 1 h in rodents [64]. Sacrifice began at 10 a.m., 5 h after lights on, and was completed within 1 h.

### 4.3. Glucose Tolerance Test

Intraperitoneal glucose (2.5 g/kg; 0.1 mL/10 g body weight) was given to 4 h fasted individually housed mice. All glucose tolerance tests began at 1 p.m., and glucose was measured in whole blood, collected from the tail vein, by glucometer (Manufacture # D2ASCCONKIT, Bayer, Leverkusen, Germany) at 0, 15, 30, 60, 90, and 120 min after glucose injection. At 15 min after glucose injection, a larger bleed (~50 μL blood) was taken from the tail vein to measure glucose-stimulated serum insulin. Blood was immediately stored on ice, and within 2 h of collection, blood was allowed to clot at room temperature for 30 min, and serum was collected after centrifugation at 3000× *g* for 30 min at 4 °C. Serum was stored at −80 °C.

### 4.4. Tissue Collection

Mice were sacrificed by decapitation after isoflurane anesthesia using the bell jar method. We collected livers and snap froze them on dry ice and trunk blood which was stored on ice. Within 2 h of collection, blood was allowed to clot at room temperature for 30 min, and serum was collected after centrifugation at 3000× *g* for 30 min at 4 °C. All tissues and serum were stored at −80 °C. Prior to analysis, frozen livers were powered using a liquid nitrogen-cooled mortar and pestle to obtain homogenous liver samples.

### 4.5. Serum Assays

Serum triglycerides (Cat. # T7531, Ponte Scientific Inc., Canton, MI, USA), glucose (Cat. # G7519, Pointe Scientific Inc., Canton, MI, USA), non-esterified fatty acids (HR Series NEFA-HR, Wako Diagnostics, Richmond, VA, USA), β-OH butyrate (Cat. # 700190, Cayman Chemicals, Pittsburg, PA, USA), and HDL and LDL/VLDL (Cat # MAK045-1KT, Sigma-Aldrich, St. Louis, MO, USA) were analyzed by colorimetric assay. Serum insulin was analyzed by enzyme-linked immunosorbent assay (ELISA; Cat. # 80-INSMSU-E01, E10, Alpco, Salem, NH, USA).

### 4.6. Liver Analyses

Whole liver mRNA was isolated from powered liver samples with TRI Reagent^®^ (Life Technologies, Grand Island, NY, USA) and purified using water-saturated butanol and ether to eliminate phenol contamination [65]. cDNA was synthesized by reverse transcription with Verso cDNA synthesis kit (Thermo Scientific, Inc., Waltham, MA, USA), and qPCR performed using SYBR 2X mastermix (Bio-Rad Laboratories, Hercules, CA, USA) and the Biorad iQ^TM^5 iCycler (Bio-Rad Laboratories, Hercules, CA, USA). Expression of β-actin (ACTβ), peroxisome proliferator-activated receptor α (PPARα), 3-hydroxy-3-methylglutaryl-CoA synthase II (HMGCS2), phosphoenolpyruvate carboxykinase (PEPCK), uncoupling protein 2 (UCP2), and carnitine palmitoyltransferase 1 (CPT1) mRNA were measured using the primer pairs previously published [3]. Expression of sterol regulatory element-binding protein I (SREBP1), sterol regulatory element-binding protein II (SREBP2), 3-hydroxy-3-methyl glutaryl CoenzymeA reductase (HMGCR), and 3-hydroxy-3-methylglutaryl-CoA synthase I (HMGCS1) were measured using the primers listed in Table 1.

LinReg PCR analysis software was used to determine the efficiency of amplification from raw output data [66]. ACTβ served as the housekeeping gene for calculating fold change in gene expression using the efficiency^−∆∆Ct^ method [67].

Total liver lipids were extracted from powered liver samples. Briefly, 10–20 mg of sample was homogenized in 100 µL PBS. 1 mL of 100% ethanol was added to each sample and agitated using a tube-holder vortex attachment for 10 min. Following 5 min of centrifugation at 16,000× *g* at 4 °C, the supernatant was transferred to a fresh tube for analysis of liver non-esterified fatty acids (HR Series NEFA-HR, Wako Diagnostics, Richmond, VA, USA) and triglycerides (Cat. # T7531, Ponte Scientific Inc., Canton, MI, USA). Liver glycogen content was quantified by a colorimetric assay as previously described [68].

Powdered liver was lysed in RIPA lysis buffer (sc-364162; Santa Cruz, Dallas, TX, USA) containing a HALT Protease and Phosphatase Inhibitor (78443; Thermo Scientific, Rockford, IL, USA). Extracted proteins were quantified by Pierce BCA Protein Assay Kit (no. 23225; Thermo Scientific, Rockford, IL, USA), and 40 µg protein was separated using 4–15% gradient Mini-Protean TGX gels (Cat.#4561085; Bio-Rad Laboratories, Inc., Hercules, CA, USA). Proteins were transferred to Polyvinylidene fluoride (PVDF) membranes using a Bio-Rad Trans-Blot Turbo (Bio-Rad). Membranes were blocked for 1 h at room temperature in 1 × TBS with 0.5% Tween 20 (TBST) and 5% nonfat dry milk (NFDM). Primary antibodies, including rabbit polyclonal anti-CPT1A (15184–1-AP; 333 µg/mL; Proteintech, Rosemont, IL, USA) were diluted 1:2000 in TBST with 5% NFDM and incubated on a rocking platform overnight at 4 °C. Membranes were washed three times for 5 min each in TBST, then F(ab’)2-Goat anti-Rabbit IgG (H + L) Cross-Adsorbed Secondary Antibody, Horseradish peroxidase (HRP; A10547; Thermo Scientific, Rockford, IL, USA) were diluted in glycerol and then diluted 1:2000 and well as β-Actin (13E5) Rabbit mAb (HRP Conjugate) (5125S, Cell Signaling, Danvers, MA, USA) diluted 1:2500 in TBST with 1% NFDM and incubated with membranes for 1 h at room temperature. Membranes were washed as before, SuperSignal West Pico PLUS Chemiluminescent Substrate (34580; Thermo Scientific, Rockford, IL, USA) was added, and the membranes were imaged with Azure biosystems c600 imager (Azure Biosystems Dublin, CA, USA) ImageJ software [69] was used for densitometry analyses.

### 4.7. Statistical Analysis

All statistical analyses were completed in SAS Enterprise Guide 7.1 (SAS Institute Inc., Cary, NC, USA). We used a mixed model ANOVA to assess the effect of genotype (GPR109a +/+ or −/−) and injection (saline or niacin) in mice that were fed or fasted. All independent variables were treated as classification variables. A Bonferroni correction was used to correct for multiple comparisons. There was no statistical difference between saline-injected GPR109a +/+ and −/− mice for any variable in either nutrition state. Accordingly, post-hoc comparisons were focused on injection (niacin or saline) within genotype. Glucose tolerance tests were analyzed by repeated measures ANOVA. Figures were created in GraphPad PRISM^®^ Version 8.2.1 for Windows (GraphPad Software, San Diego, CA, USA) and are displayed as Mean ± SEM.

## 5. Conclusions

GPR109a does not play a major role in regulating normal glucose or lipid homeostasis in either the fed or fasted state. Independent of GPR109a, niacin limits lipolysis and hepatic lipid accumulation without profound metabolic disturbances while fasting. Future work focused on understanding GPR109a independent mechanisms of niacin action will be critical to enhance the therapeutic potential of niacin-like derivatives.

## Figures and Tables

**Figure 1 ijms-22-04001-f001:**
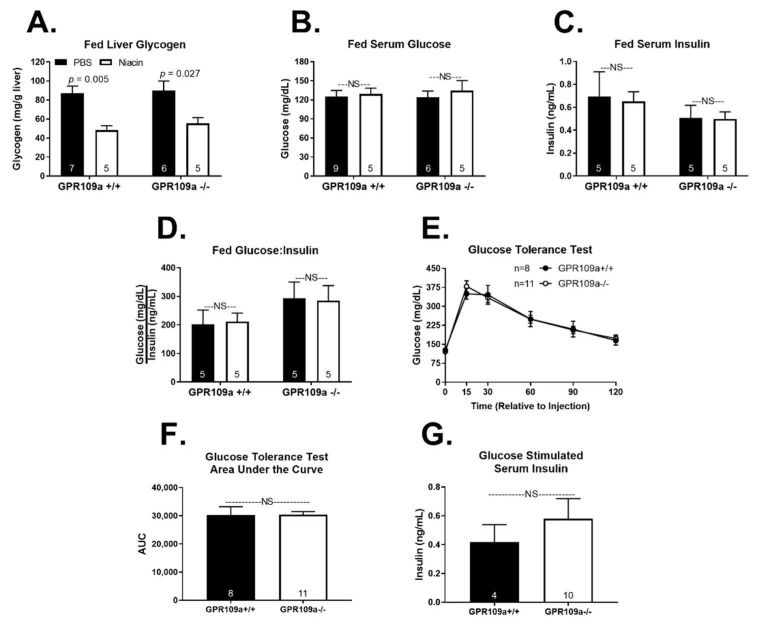
Effect of niacin on glucose homeostasis in fed GPR109a +/+ and −/− mice. Hepatic (**A**) glycogen (mg/g liver tissue), serum (**B**) glucose (mg/dL), (**C**) insulin (ng/mL), and (**D**) glucose:insulin ratio. Direct comparisons were made between injection treatment within genotype. (**E**) Glucose tolerance test in 4-h fasted mice. (**F**) Glucose tolerance test area under the curve. (**G**) Glucose stimulated serum insulin. Bars were analyzed by a two-sided unpaired *t*-test. NS = non-significant; *p* > 0.05; PBS = phosphate-buffered saline. Numbers inside bars denote the *n* per group.

**Figure 2 ijms-22-04001-f002:**
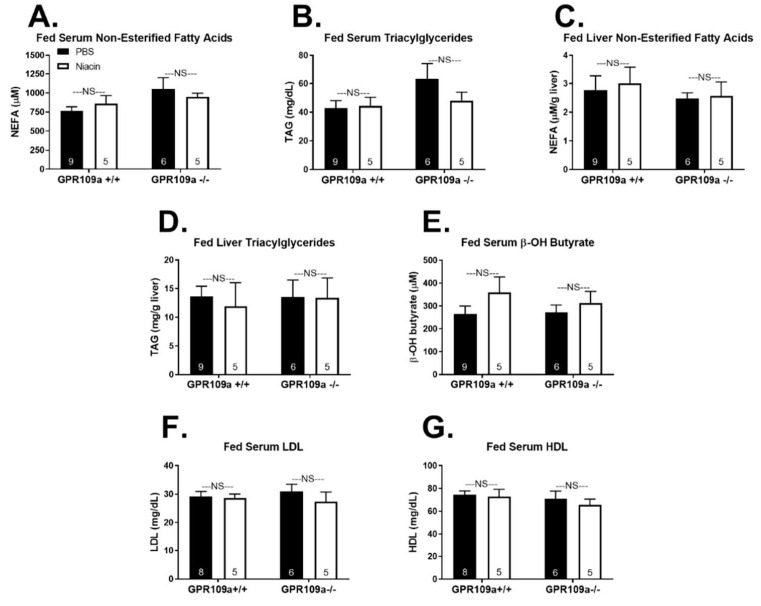
Effect of niacin on lipid and cholesterol homeostasis in fed GPR109a +/+ and −/− mice. Serum (**A**) non-esterified fatty acids (NEFA; µM) and (**B**) triacylglycerol (TAG; mg/dL). Hepatic (**C**) non-esterified fatty acids (NEFA; µmol/g liver tissue) and (**D**) triacylglycerol (TAG; mg/g liver tissue). Serum (**E**) β-OH butyrate (µM), (**F**) low-density lipoprotein (LDL; mg/dL), and (**G**) high-density lipoprotein (HDL; mg/dL). Direct comparisons were made between injection treatments within genotype. NS = non-significant; *p* > 0.05; PBS = phosphate-buffered saline. Numbers inside bars denote the *n* per group.

**Figure 3 ijms-22-04001-f003:**
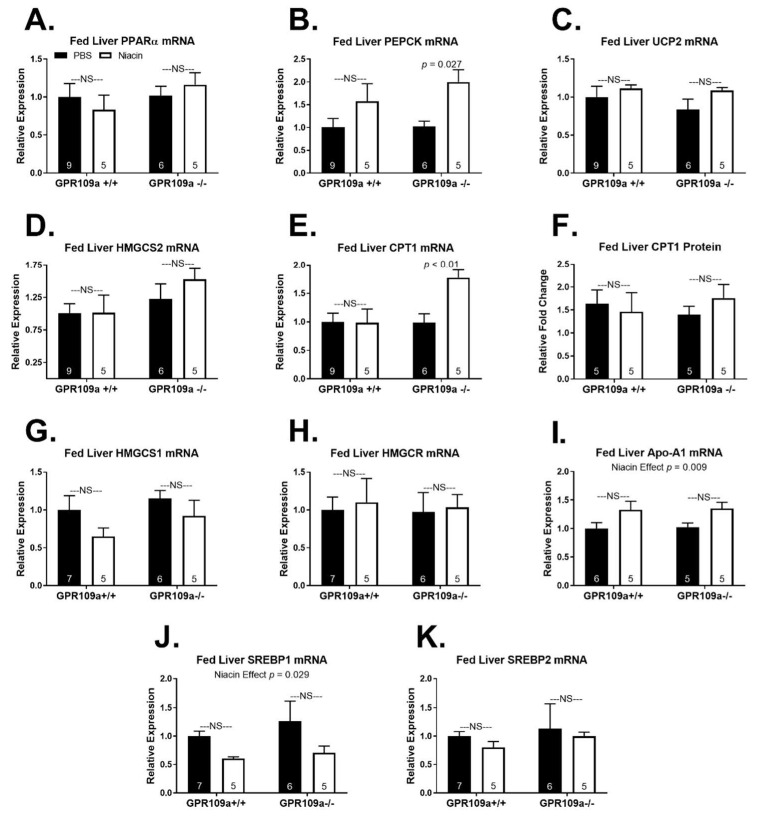
Effect of niacin on hepatic gene expression in fed GPR109a +/+ and −/− mice. Hepatic (**A**) PPARα mRNA expression, (**B**) PEPCK mRNA expression, (**C**) UCP2 mRNA expression, (**D**) HMGCS2 mRNA expression, (**E**) CPT1 mRNA expression, (**F**) CPT1 protein expression, (**G**) HMGCS1 mRNA expression, (**H**) HMGCR mRNA expression, (**I**) Apo-A1 mRNA expression, (**J**) SREBP1 mRNA expression, and (**K**) SREBP2 mRNA expression. Direct comparisons were made between injection treatments within genotypes. NS = non-significant; *p* > 0.05; PBS = phosphate-buffered saline. Numbers inside bars denote the *n* per group.

**Figure 4 ijms-22-04001-f004:**
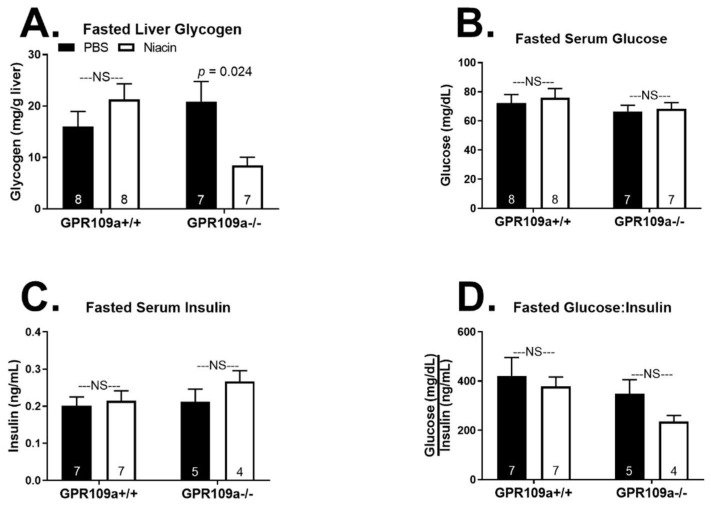
Effect of niacin on glucose homeostasis in 16 h fasted GPR109a +/+ and −/− mice. Hepatic (**A**) glycogen (mg/g liver tissue), serum (**B**) glucose (mg/dL), (**C**) insulin (ng/mL), and (**D**) glucose:insulin ratio. Direct comparisons were made between injection treatment within genotype. NS = non-significant; *p* > 0.05; PBS = phosphate buffered saline. Numbers inside bars denote the *n* per group.

**Figure 5 ijms-22-04001-f005:**
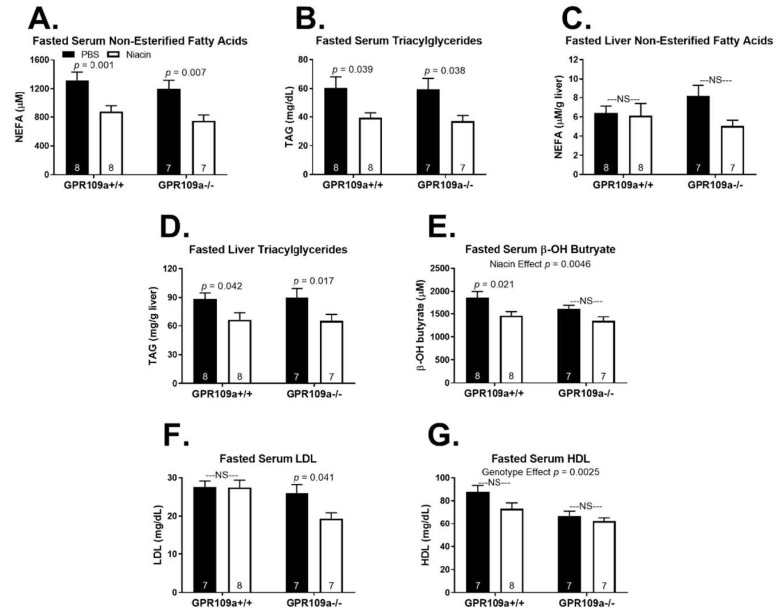
Effect of niacin on lipid and cholesterol homeostasis in 16 h fasted GPR109a +/+ and −/− mice. Serum (**A**) non-esterified fatty acids (NEFA; µM) and (**B**) triacylglycerol (TAG; mg/dL). Hepatic (**C**) non-esterified fatty acids (NEFA; µmol/g liver tissue) and (**D**) triacylglycerol (TAG; mg/g liver tissue). Serum (**E**) β-OH butyrate (µM), (**F**) low-density lipoprotein (LDL; mg/dL), and (**G**) high-density lipoprotein (HDL; mg/dL). Direct comparisons were made between injection treatments within genotypes. NS = non-significant; *p* > 0.05; PBS = phosphate-buffered saline. Numbers inside bars denote the *n* per group.

**Figure 6 ijms-22-04001-f006:**
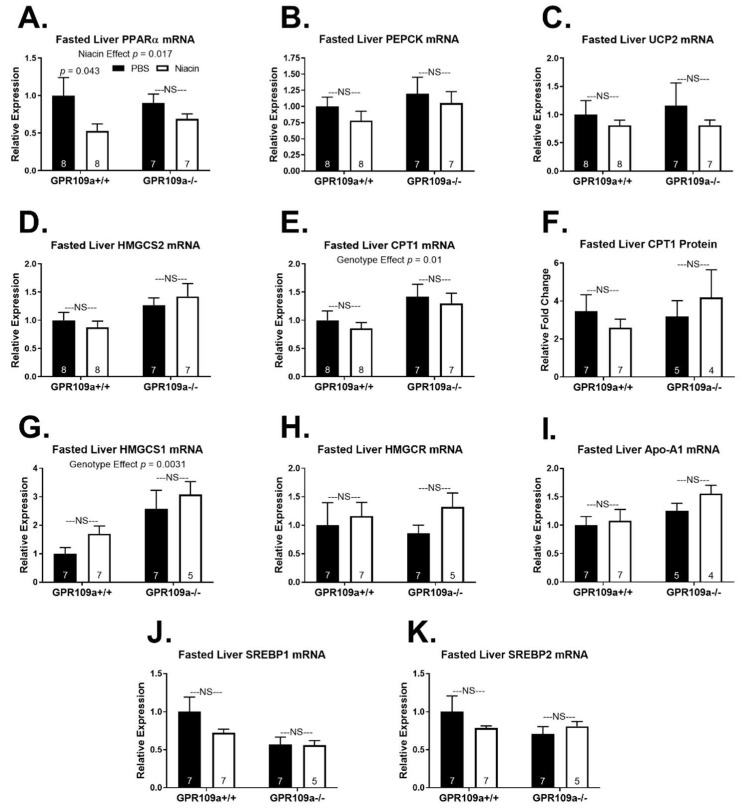
Effect of niacin on hepatic gene expression in 16 h fasted GPR109a +/+ and −/− mice. Hepatic (**A**) PPARα mRNA expression, (**B**) PEPCK mRNA expression, (**C**) UCP2 mRNA expression, (**D**) HMGCS2 mRNA expression, and (**E**) CPT1 mRNA expression, (**F**) CPT1 protein expression, (**G**) HMGCS1 mRNA expression, (**H**) HMGCR mRNA expression, (**I**) Apo-A1 mRNA expression, (**J**) SREBP1 mRNA expression, and (**K**) SREBP2 mRNA expression. Direct comparisons were made between injection treatments within genotypes. NS = non-significant; *p* > 0.05; PBS = phosphate-buffered saline. Numbers inside bars denote the *n* per group.

**Table 1 ijms-22-04001-t001:** qPCR primers.

Gene	Forward	Reverse
HMG-CoA Reductase	5′-CCTGTGGAATGCCTTGTGATTG-3′	5′-AGCCGAAGCAGCACATGAT-3′
HMG-CoA Synthase	5′-TGGCACAGTACTCACCTC-3′	5′-CCTTCATCCAAACTGTGG-3′
SREBP-1	5′-GCAGCCACCATCTAGCCTG-3′	5′-CAGCAGTGAGTCTGCCTTGAT-3′
SREBP-2	5′-GCAGCAACGGGACCATTCT-3′	5′-CCCCATGACTAAGTCCTTCAACT-3′

## Data Availability

The datasets generated and/or analyzed during the current study are available in the Mendeley data repository at the following link: http://dx.doi.org/10.17632/bz7sj7nw3w.1.

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
