# Peer review of "The Role of GPR109a Signaling in Niacin Induced Effects on Fed and Fasted Hepatic Metabolism"

_ijms, 2021, doi:10.3390/ijms22084001_

Round 1

Reviewer 1 Report

None, I am satisfied with the way answered the proposed queries

Author Response

Thank you for helping use to improve the manuscript.

Reviewer 2 Report

This is an investigation on the role of GPR109a signaling in niacin induced effects on fed and fasted glucose, lipid and cholesterol homeostasis, hepatic metabolic enzyme mRNA expression, and serum lipid, cholesterol, and ketogenic profiles in GPR109a wild type (+/+) and knockout (-/-) mice. The title and abstract are appropriate, the article is well constructed, the experiments were well conducted, and analyses were well performed.

However, the authors should clarify the following issues to avoid confusion.

  1.      Results, line 115 and Fig. 3I, line 118 and Fig. 3J, line 138 and Fig. 5E, line 143 and Fig. 5G, and so on: in the text, the authors reported statistically significant effect(s) of the niacin treatment on the various measured parameters “independent of genotype” and related them with figures that showed non-significant influence of niacin treatment within genotype. For e.g., Line 113-115: "Hepatic mRNA expression of apolipoprotein A1… was elevated by niacin treatment independent of genotype (Fig. 3I; P=0.009)." However, Fig. 3I shows that there is no significant difference in fed liver Apo-A1 mRNA between niacin treated and control mice, for both GPR109a +/+ and GPR109a -/- mice. This makes confusion. Please, consider modification of the text. Maybe it would be more clear if the authors report both results, independent of genotype and within genotype.  

2.      Line 138, P<0.005: I suggest to report exact p-value, P=0.0…

3.      Figure 5C, P<0.1: P<0.1 should be replaced by ---NS---.

4.      Abstract, line 14 (16h fasted), Results, line 130 (…effect of repeated niacin injections … after a 16 hours fast), line 133 (We report that niacin injections during the last 9 hours of a 16 hour fast decreased ... Fig. 5A-5B), Figures 4, 5, 6 (Effect of niacin … in 16h fasted mice), Discussion, line 179/180 (We investigated the role of GPR109a expression in acute (9h) niacin mediated changes…): It seems confusing.

5.      Line 117, “Niacin to decreased fed state hepatic…”: a verb is missing.

6.      Line 240, “One possible explanation for this discrepancy is the timing between niacin exposure and NEFA quantification”: The dose of niacin used in this study differed from the doses used in the studies of Tunaru at al (Ref. 1) and Lauring  et al. (Ref. 15). Can this also contribute to different results?

7.      Materials and Methods, Injection Studies: Why is niacin administered at a dose of 0.8 mmol/kg?

8.      Line 371: It should be “2-ΔΔCT method” i.e.    (https://www.sciencedirect.com/science/article/pii/S1046202301912629)

9.      Line 387, “weas” probably should be “were”.

10.  Serum Assays: Methods for cholesterol, HDL and LDL are missing?

11.  Statistical Analysis, a Bonferroni correction was used: What is the new p-value? Please, check a significance of the difference between the groups for the presented results.

12.  Ref. 12: The authors are: AIM-HIGH Investigators; links on the paper: https://www.nejm.org/doi/pdf/10.1056/NEJMoa1107579?articleTools=true; https://www.nejm.org/doi/suppl/10.1056/NEJMoa1107579/suppl_file/nejmoa1107579_appendix.pdf (writing group); https://pubmed.ncbi.nlm.nih.gov/22085343/

13. References should be prepared according to Instructions for Authors.

Author Response

Thank you for helping us to improve this manuscript.  We have attached a word file with a direct response to each point raised.

This manuscript is a resubmission of an earlier submission. The following is a list of the peer review reports and author responses from that submission.

Round 1

Reviewer 1 Report

Geisler & Renquist investigate the role of GPR109a signalling in niacin induced effects on fed and fasted hepatic metabolism. The authors compare liver and the niacin response in GPR109a-/- to littermate GPR109a +/+ controls.

Liver and plasma metabolites and mRNA are quantified following acute niacin treatment at a single timepoint (9h) in a fed and fasted state. The introduction clearly introduces relevant studies on GPR109a. Relevant research is cited. Methods are clearly stated. The ketone body b-hydroxybutyrate is the endogenous ligand for GPR109a, explaining the rationale for measuring niacin effects in a fed and fasted state.

The following would improve the effectiveness of the study:

The authors explain that niacin therapy is useful for its role in lowering circulating TAG and VLDL and raising HDL concentrations in patients with dyslipidemia and indicate the aim of this work is to maximize the clinical applications of niacin therapy. Although plasma and liver TAG are measured, measurement of cholesterol and lipoproteins are lacking. Analysis of the cholesterol synthesis pathway regulation at the protein level and as well as SREBP activity would be helpful.

Metabolic measurements are limited to one timepoint after acute niacin administration. Since niacin treatment for hypercholesterolemeia is chronic, measurements of chronic treatment would appear to be more useful. Justification for the one timepoint and acute administration rationale are lacking. A timecourse would be useful for this.

It is unclear if the study is investigating the liver-specific role of GPR109a or the effect of global GPR109a KO on the liver. Since adipocytes are a major site of GPR109a action, and there is substantial crossstalk between adipose and liver, the absence of data on adipose tissue make it difficult to dissect GPR109a effects. Tissue specific KO would be more useful. Evidence of GPR109a expression in the liver should be included.

Niacin induced decrease in liver glycogen was observed only in the absence of GPR109a. This was a significant difference between the KO and control. In the discussion the authors indicate that the mechanism for glycogen loss is infact independent of GPR109a and cite alternative mechanisms for niacin regulation of liver glycogenolysis via glycogen phosphoyralse acetylation. An alternative explanation for the observed results is required.

In the discussion the authors cite previous data that has implicated GPR109a in insulin secretion and insulin resistance and state that their data showing no effect on glucose tolerance or glucose stimulated serum insulin (line 151-156). Addition of possible explanations for inconsistency with previous data such as specific compensatory mechanisms would be helpful here.

Given the variability of the data, power analysis should be undertaken to determine the appropriate number of mice required for each experiment.

The data are insufficient to conclude that GPR109a is not essential for the metabolic response to the ketogenic state or the pharmacological benefits associated with niacin.

Reviewer 2 Report

The manuscript submitted by CE Geisler and BJ Renquist addresses and interested subject, but it is at a preliminary stage in its elaboration. Several key points arise after reading the manuscript that should be properly worked out:

. What is it the real expression of GPR109, at the mRNA and protein level, in the knockout mice at the fed and fasted states

. Most of the conclusions are derived mainly from mRNA quantitation of key metabolic genes. Data about protein expression, enzyme activity (i.e. fatty acid beta-oxidation driven by PPAR alpha activation) should be obtained and considered

. Niacin seems to do not affect glucose homeostasis in the fed as well as in the fasted state. OTT is performed by intraperitoneal injection, avoiding the participation of GI peptides and metabolic signals that are activated when glucose is administered orally, its physiological way to enter the organism. This subject should be given more consideration

. Changes in metabolic parameters between the fed and fasted states in the wild type and knockout conditions should be properly addressed and commented.